# The development of an alternative growth chart for estimated fetal weight in the absence of ultrasound: Application in Indonesia

Dewi Anggraini[1¤a]*, Mali Abdollahian[2©¤b], Kaye Marion[2©]

1 Study Program of Statistics, Faculty of Mathematics and Natural Sciences, Lambung Mangkurat University, Banjarbaru, South Kalimantan, Indonesia, 2 School of Science, College of Science, Engineering, and Health, RMIT University, Melbourne, Victoria, Australia

© These authors contributed equally to this work.
¤a Current address: Study Program of Statistics, Faculty of Mathematics and Natural Sciences, Lambung Mangkurat University, Banjarbaru, South Kalimantan, Indonesia
¤b Current address: School of Science, College of Science, Engineering, and Health, RMIT University, Melbourne, Victoria, Australia
* dewi.anggraini@ulm.ac.id

**Data Availability Statement:** All relevant data are within the manuscript and its Supporting Information files.

## Abstract

A fetal growth chart is a vital tool for assessing fetal risk during pregnancy. Since fetal weight cannot be directly measured, its reliable estimation at different stages of pregnancy has become an essential issue in obstetrics and gynecology and one of the critical elements in developing a fetal growth chart for estimated fetal weight. In Indonesia, however, a reliable model and data for fetal weight estimation remain challenging, and this causes the absence of a standard fetal growth chart in antenatal care practices. This study has reviewed and evaluated the efficacy of the prediction models used to develop the most prominent growth charts for estimated fetal weight. The study also has discussed the potential challenges when such surveillance tools are utilized in low resource settings. The study, then, has proposed an alternative model based only on maternal fundal height to estimate fetal weight. Finally, the study has developed an alternative growth chart and assessed its capability in detecting abnormal patterns of fetal growth during pregnancy. Prospective data from twenty selected primary health centers in South Kalimantan, Indonesia, were used for the proposed model validation, the comparison task, and the alternative growth chart development using both descriptive and inferential statistics. Results show that limited access to individual fetal biometric characteristics and low-quality data on personal maternal and neonatal characteristics make the existing fetal growth charts less applicable in the local setting. The proposed model based only on maternal fundal height has a comparable ability in predicting fetal weight with less error than the existing models. The results have shown that the developed chart based on the proposed model can effectively detect signs of abnormality, between 20 and 41 weeks, among low birth weight babies in the absence of ultrasound. Consequently, the developed chart would improve the quality of fetal risk assessment during pregnancy and reduce the risk of adverse neonatal outcomes.

**Funding:** The authors received no specific funding for this work. However, the Australian Agency for International Development (AusAID) has granted DA's PhD scholarship in Mathematical Sciences at the School of Science, RMIT University, Melbourne, Australia. This analysis is part of DA's thesis.

**Competing interests:** The authors have declared that no competing interests exist.

**Abbreviations:** ANC, Antenatal care; GA, Gestational age; EFW, Estimated fetal weight; MCH, Maternal and child health; FH, Fundal height; LBW, Low birth weight; ABW, Actual birth weight; LSL, Lower specification limit; USL, Upper specification limit; AC, Abdominal circumference; HC, Head circumference; GAMLSS, Generalized additive models for location, scale, and shape; WHO, World health organization; FL, Femur length; MUAC, Middle upper arm circumference; BMI, Body mass index; ME, Mean prediction error; MPE, Mean percentage error; MAPE, Mean absolute percentage error; MEDPE, Median percentage error; MEDAPE, Median absolute percentage error; BPD, Biparietal diameter.

## Introduction

Fetal growth is one of the critical indicators for assessing fetal risk during pregnancy. The assessment of fetal growth has been well documented as one of the objectives of antenatal care (ANC) to reduce the risk of neonatal mortality and morbidity [1–3]. Monitoring the intrauterine development of the fetus at different stages of pregnancy, even at earlier gestational ages (GAs), is vital for early detection of growth abnormalities [4]. In this way, delays in making informed referrals, decisions, and interventions can be minimized to ensure safe delivery and a positive pregnancy outcome, particularly crucial for those who are living in rural areas or settings with limited health resources.

Birth weight is closely associated with fetal growth [5]. The measurement of fetal weight during pregnancy is used to estimate the expected (normal) birth weight [6]. Since fetal weight cannot be directly measured, its reliable estimation at different stages of pregnancy has become one of the important issues in obstetrics and gynecology, and one of the key elements in developing a fetal growth chart for the estimated fetal weight (EFW).

The provision of a fetal growth chart for EFW has been identified as one of the fundamental preventive actions to promote healthy human development and to prevent the risk of common non-communicable diseases in later life [5, 7]. The impact of utilizing the growth chart in clinical practice has been investigated [2, 4, 8–10]. The results show a significant improvement in the detection of abnormal growth and a remarkable reduction in unnecessary referrals.

Access to routine data collection of EFW at a given GA is required to develop an evidence-based fetal growth chart. In Indonesia, however, such data collection remains challenging and is one of the reasons for the absence of a standard fetal growth chart [11–15]. There is no fetal growth chart for EFW available in the current ANC practice of Indonesian primary health care centers. Its absence is particularly noticeable in the maternal and child health (MCH) booklet (Buku KIA) [16], which is nationally recognized as one of the vital profile monitoring records during pregnancy.

This study aims to fill the gap by, first, reviewing and evaluating the efficacy of the prediction models used to develop the most prominent growth charts for EFW. Then, discussing the potential challenges when such surveillance tools are utilized in low resource settings, proposing an alternative growth chart for EFW in the absence of ultrasound, and analyzing the capability of the proposed chart in detecting abnormal patterns of fetal growth during pregnancy.

## Methods

### Review of existing fetal growth charts for estimated fetal weight and the applicability in low resource settings

A purposive literature review was performed for searching original research articles on the most prominent growth charts for EFW. The selection criteria for the research articles used in this study were based on the sizes of the study population, and the ways fetal growth charts were developed, including both customized and standard approaches. The review and evaluation were explicitly focused on the statistical models used to predict fetal weight and the potential challenges when the charts are utilized in Indonesian health care centers where advanced health equipment and facilities are not always available, or indeed necessary. From the chosen literature, the significant characteristics for each model were identified and investigated whether measurements of these characteristics might be available in the rural settings of Indonesia.

## Local evaluation of ultrasound-based prediction models used in the development of existing growth charts for estimated fetal weight and the comparison with the proposed clinical model

A prospective cohort study, between June 1, 2016, and June 30, 2017, was carried out in South Kalimantan, Indonesia. The detailed procedures have been provided in previous publications [12, 15]. The prospective data, both clinical and ultrasound information, were used to validate the existing ultrasonic-based and proposed clinical-based models for estimating fetal weight. The data were obtained from one of the twenty selected primary health care centers that have access to both ultrasound measurement of fetal biometric characteristics and clinical assessment of fundal height (FH) at a given GA. All measurements (from the beginning of pregnancy to delivery) were performed by the dedicated midwife who runs the center. This midwife has had both ultrasound and scientific and technical training [12, 15]. The selected midwife has experienced and been involved in both the provision and delivery of ANC services for $\geq$ five years (actually 22 years).

During the period of study, 30 pregnant women regularly attended ANC and gave birth to the selected center. One (3%) pregnant woman was excluded from the analysis due to low birth weight (LBW) newborn ($<$ 2500 g). Ten women (33%) were excluded from the validation analysis due to unmet inclusion criteria S1 Fig.

A comparison study was carried out between the existing and the proposed models using the de-identified prospective data. The existing models were developed based on data recorded within one week of delivery [17, 18] and within 14 days of the last ultrasound scan [19] while the proposed model:

$$\mathrm{EFW(g)} = 109.16\mathrm{FH} - 272 \tag{1}$$

was developed based on FH measurement recorded between 35 and 41 weeks (before delivery) and actual birth weight. The model in Eq (1) was the improvement of the clinical model published in Anggraini, et al. [6], which was developed using recorded estimated weight based on the existing Johnson-Toshach formula [20].

Several accuracy measures used in this study, such as mean prediction error (ME), mean percentage prediction error (MPE), mean absolute percentage prediction error (MAPE), and median absolute percentage prediction error (MEDAPE). The number of estimates within 10% and 20% of ABWs (%) is also used to measure the prediction accuracy [6]. In this study, however, a two-sample T-test and a two-sample F-test were added to investigate if there is a significant difference in mean and variance of prediction errors between the existing and proposed models. The prediction accuracy of the existing and proposed models was assessed using the prospective data of 19 pregnant women recorded at different stages of pregnancy: between 16 and 38 weeks (during pregnancy) and between 33 and 40 weeks (before or at delivery).

## The development of an alternative growth chart for estimated fetal weight in the absence of ultrasound

This study used information from 435 participating women enrolled in twenty selected primary health care centers, including the prospective cohort study explained in the previous section. Of these, 33 participating women (7.6%) were excluded from the study due to LBW (n = 16), no gender and birth weight information (n = 1), no measurements taken during pregnancy (n = 2), and no information on GA and FH (FH $<$ 12 cm and FH $>$ 38 cm [1], n = 14). Therefore, a total of 402 participating women provided the basis for the development of a true average growth curve based on a weight prediction model that does not use ultrasound data S2 Fig.

In this study, a Bernoulli distribution with the event probability (p) of 70% was used to randomly divide 1,408 FH measurements of 402 pregnant women into two data sets. The first data set consists of 989 EFW measurements of 385 pregnant women (training data). The second data set consists of 419 (30%) EFW measurements of 282 pregnant women (testing data).

Since this study aims to develop an alternative fetal growth chart for EFW that does not use an ultrasound-based weight prediction model, the proposed clinical model, based only on FH measurement [Eq (1)], was used to estimate the fetal weight at a given GA. The estimated values of fetal weight were then plotted against GA to develop the optimal model of weight prediction based on GA using the curve fitting option in SPSS 23. The idea of this model development was to create the percentile limits (profile limits) for EFW. Regression analysis using curve estimation was used to develop the relationship between EFW (estimated using the proposed clinical model) and GA.

Diagnostic tests of residuals were carried out using the Ryan-Joiner or Shapiro-Wilk analysis to compare the sample distribution to a normal curve [21]. Also, a probability plot and histogram with normal curve fit were created to inspect the distribution of residuals visually. The testing data set was used to validate and assess the efficacy of the proposed fetal weight prediction models based on GA. The residual was calculated based on the difference between the actual birth weight (ABW) and the EFW-GA. Multiple comparisons were then carried out between the proposed EFW-GA model and two existing ultrasound-based EFW-GA models [22, 23] to select the most effective model in predicting EFW based on GA. The comparisons were presented in two periods: during pregnancy (13–42 weeks) and before or at delivery (32–42 weeks).

This study used the research procedures proposed by Mikolajczyk, et al. [24] to develop an alternative fetal growth chart for EFW. This existing method has been recommended by Gardosi [25] to allow different countries (where reliable population data are not available) to use their own population information to create the references rather than using reference curves based on diverse populations. However, the existing method adapted an ultrasound-based formula proposed by Hadlock, et al. [22] to predict EFW based on GA and create the profile limits for EFW.

In this research, the proposed clinical quadratic EFW-GA model:

$$EFW = 137.173GA - 1.035GA^2 - 675.199 \tag{2}$$

was used to develop the chart. The actual values of birth weight and the predicted values of fetal weight at term pregnancy (between 37 and 41 weeks) (using the value of the 50th percentile weight for GA) were first compared. Then, the proposed alternative fetal growth chart for EFW was implemented among 282 normal newborns (testing data) and 16 LBW newborns identified among 435 singleton live newborns in the prospective study S2 Fig. The capability of the proposed chart in detecting normal and unusual patterns of fetal growth was analyzed by calculating the number of cases and its percentage that falls below the lower specification limit (LSL), i.e., the 10th percentile, between the 10th and 90th percentiles and above the upper specification limit (USL), i.e., the 90th percentile [9].

### Ethics approval and consent to participate

This research was conducted using retrospective (past and current) and prospective (longitudinal) data collected from Indonesia. Since the study dealt with personal data, two main ethics clearances were obtained:

1. The Medical Research Ethics Committee, Medical Faculty, University of Lambung Mangkurat, Banjarmasin, South Kalimantan, Indonesia, on March 10, 2016, with registration

number 018/KEPK-FK UNLAM/EC/III/2016. Based on the letter-number 019/KEPK-FK UNLAM/EC/III/2016, the validity period of the ethical clearance is from March 10, 2016, until March 2, 2019, or during the time that the research takes place. Permission to access de-identified secondary data in the pregnancy register available at the selected primary health care centers was also granted under this ethics approval.

2. The Science, Engineering, and Health College Human Ethics Advisory Network, RMIT University, Melbourne, Australia, on March 16, 2016, with registration number ASEHAPP 19-16/RM No: 19974.

Research permissions were also obtained from the Indonesian national, provincial, and local governments and the provincial health department. Information about the confidential nature of the project and consent forms (written in Bahasa Indonesia and English) for recruitment to the study was given to the selected midwives, who all agreed to participate. These forms were also provided for the pregnant women who participated in the prospective research through the representative midwives. All research participants have also signed and returned the consent forms to the principal researcher.

## Results

### Review of existing growth charts for estimated fetal weight and evaluation of their applicability in low resource settings

Five original research articles of the most prominent studies on fetal growth charts for EFW were selected: customized fetal growth chart studies (2 articles) and standard fetal growth chart studies (3 articles). The review and evaluation of these studies were summarized and provided in the S1 Table.

The first two fetal growth charts for EFW, proposed by Gardosi et al. [26, 27], were developed by considering individual characteristics that significantly influence fetal growth. These widely used charts are referred to as customized fetal growth charts. Both charts were derived from a retrospective study among the British population. Access to ultrasound measurements of fetal biometric characteristics and the minimum database of maternal and fetal characteristics that have a potential impact on fetal growth was required to develop the charts. Also, information on maternal weight at the first visit, height, parity, ethnicity, and gender of fetus/neonate (if known) was needed. These characteristics were used to adjust the range of EFW for an individual pregnant woman.

The remaining selected fetal growth charts for EFW were derived from either retrospective or prospective studies across countries with homogenous or heterogeneous populations but healthy, well-nourished, and of low risk of adverse pregnancy outcomes at both population and individual levels. Unlike customized fetal growth charts, these charts were developed without considering individual variability. These charts are referred to as standard fetal growth charts.

The first standard growth chart, proposed by Mikolajczyk, et al. [24], is referred to as a global (generic) reference for fetal weight and birth weight percentiles because it combines fetal weight estimation with the notion of proportionality, as proposed by Gardosi, et al. [26]. The profile limits for EFW were developed based on the regression analysis between EFW and GA using Hadlock et al.'s [22] model. The second standard growth chart, proposed by the Intergrowth 21$^{st}$ project [19], was best developed using the fetal weight estimation based on fetal abdominal circumference (AC) and fetal head circumference (HC) measured between 0 and 14 days before delivery. The profile limits for EFW were developed using the Generalized Additive Models for Location, Scale, and Shape (GAMLSS) framework. The last standard

growth chart, put forward by the World Health Organization (WHO) [5], was developed using the fetal weight estimation model proposed by Hadlock et al. [22] to create the profile limits for monitoring the change of EFW in terms of GA. However, the selected formula for EFW was based on Hadlock, et al.'s [28] that combines three (instead of four) fetal characteristics [HC, AC, and femur length (FL)] measured by ultrasound within 3–7 days of delivery.

## Local evaluation of prediction models used in the development of existing growth charts for estimated fetal weight and the comparison with the proposed model

**General information on the study population.** Descriptive statistics of the study population (n = 19) are presented in the S2 Table. All pregnant women included in this study followed the minimum recommendation of ANC visits (8 visits). Overall, the pregnant women were between 23 and 32 years old (73.7%), well-nourished with middle-upper arm circumference (MUAC) $\geq$ 23.5 cm (84.2%), multiparous (68.4%) and had normal body mass index (BMI) at the first visit of ANC (between 18.5 and 24.9 kg/m$^2$) (73.7%). Of these 19 women, 16 (84.2%) delivered the baby through spontaneous (normal) mode, between 37 and 44 weeks, and 15 (79%) were assisted by midwives.

**Local validation: Using Indonesian ANC data recorded between 16 and 38 weeks (during pregnancy) and between 33 and 40 weeks (before or at delivery).** The prediction accuracy of the existing and proposed models was assessed using the prospective data of 19 pregnant women recorded at different stages of pregnancy: between 16 and 38 weeks and between 33 and 40 weeks. The results are presented in Tables 1 and 2, respectively.

Tables 1 and 2 show the accuracy comparison between the existing and proposed models. The tables have several measures of prediction errors, such as the mean prediction errors (MEs), the mean percentage errors (MPEs), the mean absolute percentage errors (MAPEs), the median percentage errors (MEDPE), and the median absolute percentage errors (MEDAPEs). It can be seen from the tables that the proposed model [Equation (1)] produced

**Table 1. Accuracy of the existing and proposed models (16–38 weeks).**

| Prediction error (ABW–EFWp) | Mean (ME) (g) | Mean percentage (MPE) (%) | Mean absolute percentage (MAPE) (%) | Median percentage (MEDPE) (%) | Median absolute percentage (MEDAPE) (%) | Number of estimates within 10% of ABWs (%) | Number of estimates within 20% of ABWs (%) |
|---|---|---|---|---|---|---|---|
| Number of pregnant women = 19 (53 observations) | | | | | | | |
| **Proposed clinical model** | | | | | | | |
| Dewi, Mali, and Kaye (2019): FH | 1,163.36 | 36.17 | 37.46 | 34.78 | 34.78 | 4 | 23 |
| **Existing ultrasound models** | | | | | | | |
| Campbell and Wilkin (1975): AC | 1,735.93 | 54.50 | 55.47 | 55.12 | 55.12 | 4 | 11 |
| Hadlock (1985) I: AC and FL | 1,861.86 | 58.69 | 58.69 | 59.92 | 59.92 | 6 | 8 |
| Hadlock (1985) II: AC, BPD, and FL | 1,849.20 | 58.30 | 58.36 | 60.54 | 60.54 | 6 | 8 |
| Hadlock (1985) III: AC, HC, and FL | 1,890.65 | 59.65 | 59.66 | 60.77 | 60.77 | 4 | 6 |
| Hadlock (1985) IV: AC, BPD, HC, and FL | 1,875.13 | 59.15 | 59.21 | 60.87 | 60.87 | 4 | 6 |
| Stirnemann (2017): HC and AC | 1,888.09 | 59.49 | 59.58 | 61.96 | 61.96 | 2 | 6 |

**Table 2. Accuracy of the existing and proposed models (33–40 weeks).**

| Prediction error (ABW–EFWp) | Mean (ME) (g) | Mean percentage (MPE) (%) | Mean absolute percentage (MAPE) (%) | Median percentage (MEDPE) (%) | Median absolute percentage (MEDAPE) (%) | Number of estimates within 10% of ABWs (%) | Number of estimates within 20% of ABWs (%) |
|---|---|---|---|---|---|---|---|
| **Number of pregnant women = 19 (19 observations)** | | | | | | | |
| **Proposed clinical model** | | | | | | | |
| Dewi, Mali, and Kaye (2019): FH | 235.09 | 7.08 | 11.44 | 10.52 | 11.09 | 42 | 84 |
| **Existing ultrasound models** | | | | | | | |
| Campbell and Wilkin (1975): AC | 272.18 | 8.46 | 11.69 | 11.61 | 12.03 | 42 | 89 |
| Hadlock (1985) I: AC and FL | 269.76 | 8.28 | 15.24 | 12.34 | 15.06 | 26 | 84 |
| Hadlock (1985) II: AC, BPD, and FL | 247.37 | 7.58 | 14.86 | 11.00 | 12.45 | 26 | 79 |
| Hadlock (1985) III: AC, HC, and FL | 338.45 | 10.50 | 15.36 | 13.59 | 14.72 | 26 | 84 |
| Hadlock (1985) IV: AC, BPD, HC, and FL | 299.07 | 9.24 | 15.12 | 12.98 | 13.07 | 26 | 79 |
| Stirnemann (2017): HC and AC | 503.58 | 16.04 | 17.61 | 18.55 | 18.55 | 21 | 58 |

smaller errors in predicting actual birth weight (ABW) than the existing models between 16 and 38 weeks and between 33 and 40 weeks.

Tables 1 and 2 also show a smaller number of predictions falling within 10% and 20% of ABWs for all models during pregnancy (16–38 weeks). However, the proposed model is significantly comparable (84%) to the Hadlock, et al. 's [18] models I and III in predicting fetal weight. The proposed model is even better than the Hadlock models II and IV (79%) and the Stirnemann et al. 's [19] (58%) when the estimation of fetal weight was made before or at delivery (between 33 and 40 weeks) with the proportion of predicted birth weights falling within 20% of ABWs. Although Campbell and Wilkin's [17] model was slightly more capable (89%), it has somewhat higher prediction errors compared to the proposed model.

The visual comparison of the MEDAPEs can be seen in Figs 1 and 2.

From Figs 1 and 2, it can be concluded that the proposed model is significantly more capable (less error) than the existing models in predicting fetal weight during pregnancy, even at earlier GAs. It also can be seen that the prediction errors are less pronounced when the pregnancy reaches advanced GAs.

The results show that, between 16 and 38 weeks, there is a significant difference in variances and means of prediction errors between the existing and proposed models during pregnancy (p-value < 0.05), except for the variation between the proposed model and Stirnemann et al. 's [19] model (p-value = 0.128) S3 and S4 Tables. There is no significant difference in variances and means of prediction errors, between 33 and 40 weeks, between the existing and proposed models (p-value > 0.05), except for the mean between the proposed model and Stirnemann et al. 's [19] model (p-value = 0.032).

**Local evaluation: Assessment of two prominent existing growth charts (the Intergrowth 21st Project and WHO) for estimated fetal weight.** The existing fetal growth charts (the Intergrowth 21st project and the WHO) were applied to the prospective data (19 pregnant women with 53 fetal measurements by ultrasound). The results are shown in Figs 3 and 4, with the red line being the 10th percentile, green the 50th, and orange the 90th.

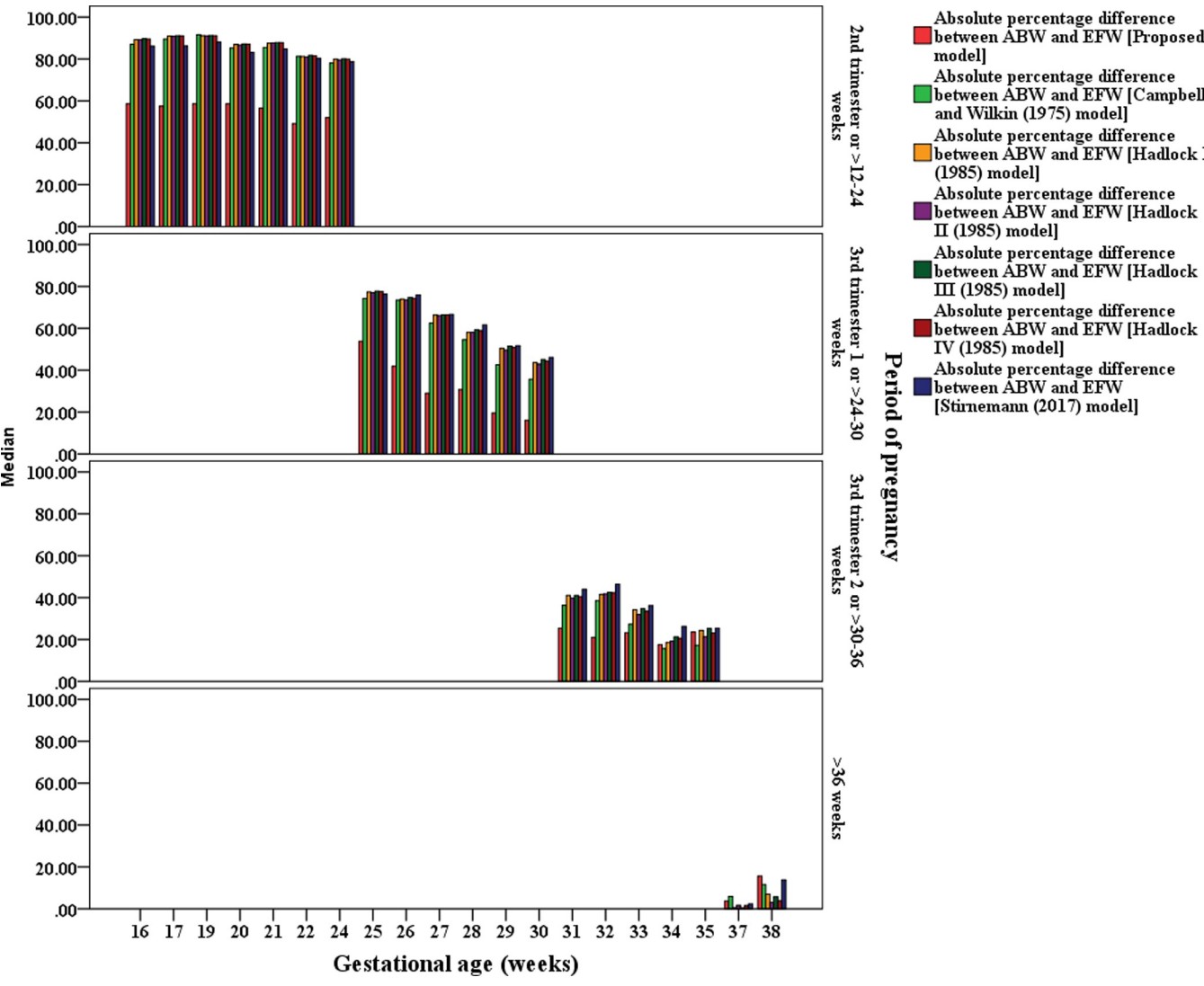

**Fig 1. MEDAPEs of the proposed and existing models (16–38 weeks).**

It can be seen from Figs 3 and 4 that both charts are comparable in fitting the Indonesian data when fetal biometric measurements using ultrasound are available. The WHO chart (Fig 4) fits the local population more effectively than the one proposed by the Intergrowth 21st Project (Fig 3). This can be seen from the distribution of raw observations falling within the 10th, 50th, and 90th percentiles.

## The development of an alternative growth chart for estimated fetal weight in the absence of ultrasound

**Description of the study population.** All 435 pregnant women included in this study followed the minimum recommendation of ANC visits (average of 6 visits) with a range between 1 and 14 visits, and only clinical ANC data were used. Overall, the pregnant women were between 23 and 32 years old (61.4%), well-nourished (37.5%) with MUAC $\geq$ 23.5 cm, multiparous (72.0%), and had normal BMI at the first visit of ANC (between 18.5 and 24.9 kg/m$^2$) (71.3%). Of these 435 women, 391 (89.9%) delivered the baby at term pregnancy, 400 (92.0%)

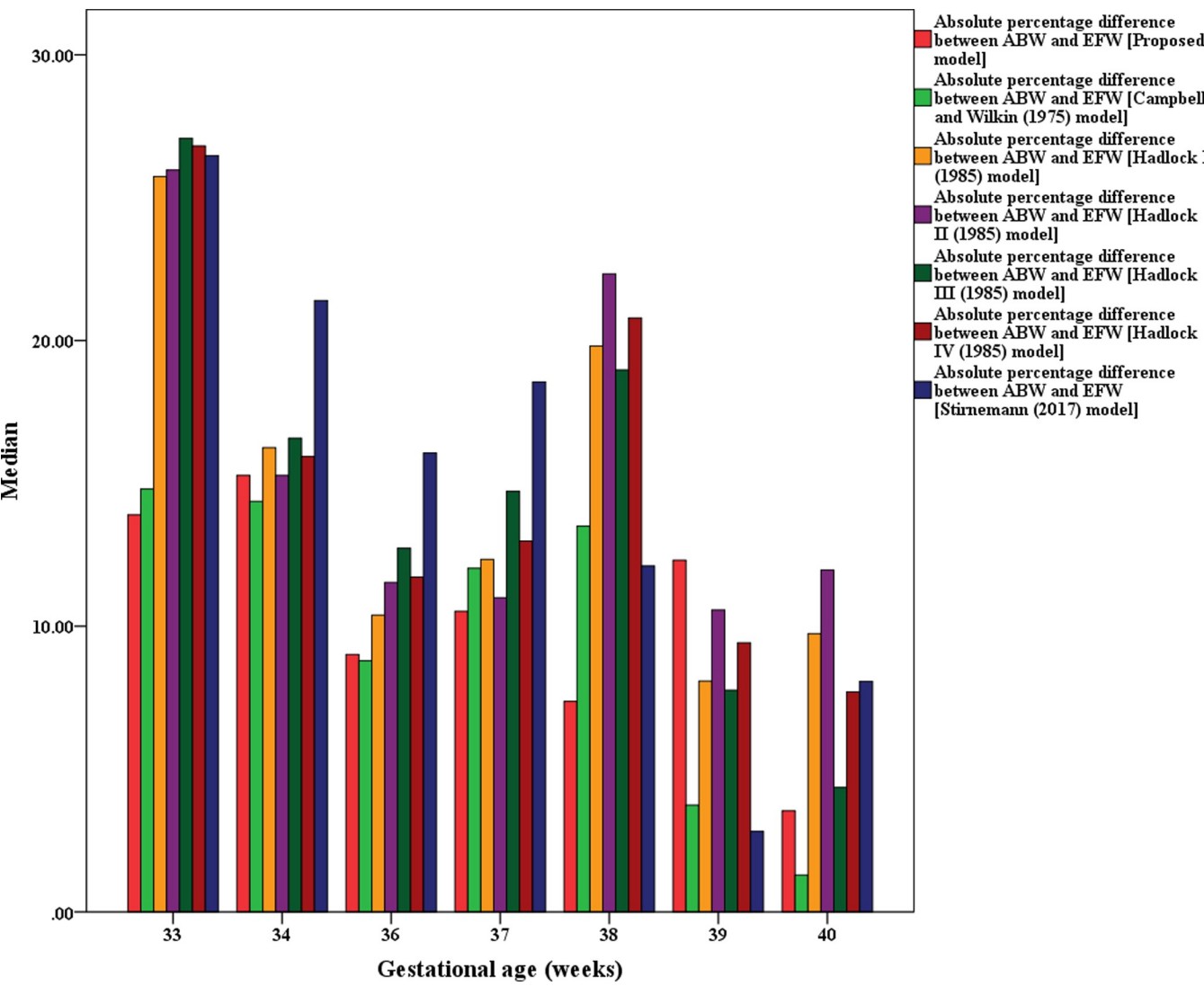

**Fig 2. MEDAPEs of the proposed and existing models (33–40 weeks).**

delivered the baby with spontaneous normal mode, 34 (7.8%) delivered preterm (< 37 completed weeks of gestation), 227 (52.2%) delivered male newborns, 200 (46.0%) delivered female newborns, and 356 (81.8%) were assisted by midwives S5 Table.

**Efficacy assessment and comparison between the existing ultrasound-based and proposed clinical-based weight prediction models based on gestational age.** The quadratic model [Eq(2)] is the best fit clinical model (it has the highest values of R = 0.851). The $R^2$-adjusted (72.4%) shows a significant correlation between EFW and GA. This model also has the least standard error of prediction (309.0 g) S6 Table. Fig 5 shows a scatter plot between EFW (Y) and GA (X) for the best fit quadratic model together with its 95% confidence intervals.

Diagnostic tests of residuals were carried out, and the results are provided in the S7 Table. Based on the Ryan-Joiner or Shapiro Wilk test, it was concluded that the residuals of all the proposed EFW-GA models were roughly symmetric, particularly for the linear model (p-value = 0.025 > 1%).

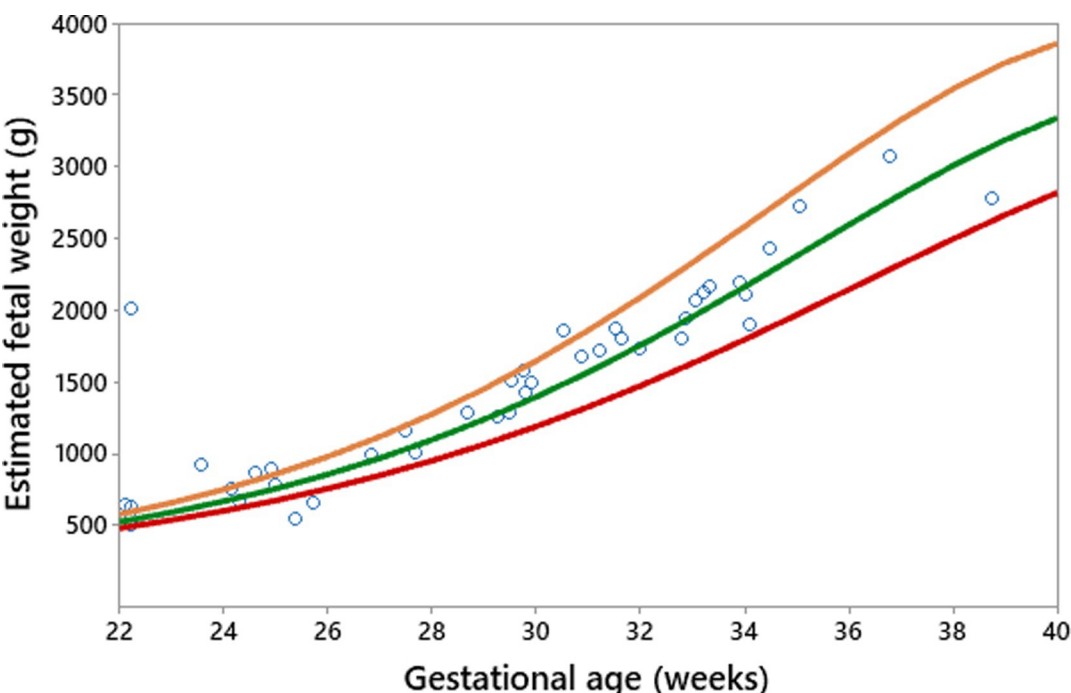

**Fig 3. Intergrowth 21st Project fetal growth chart applied to an Indonesian population.**

The prediction accuracy between the proposed clinical-based EFW-GA model and the existing ultrasound-based EFW-GA models proposed by Hadlock et al. [22] and Sotiriadis,

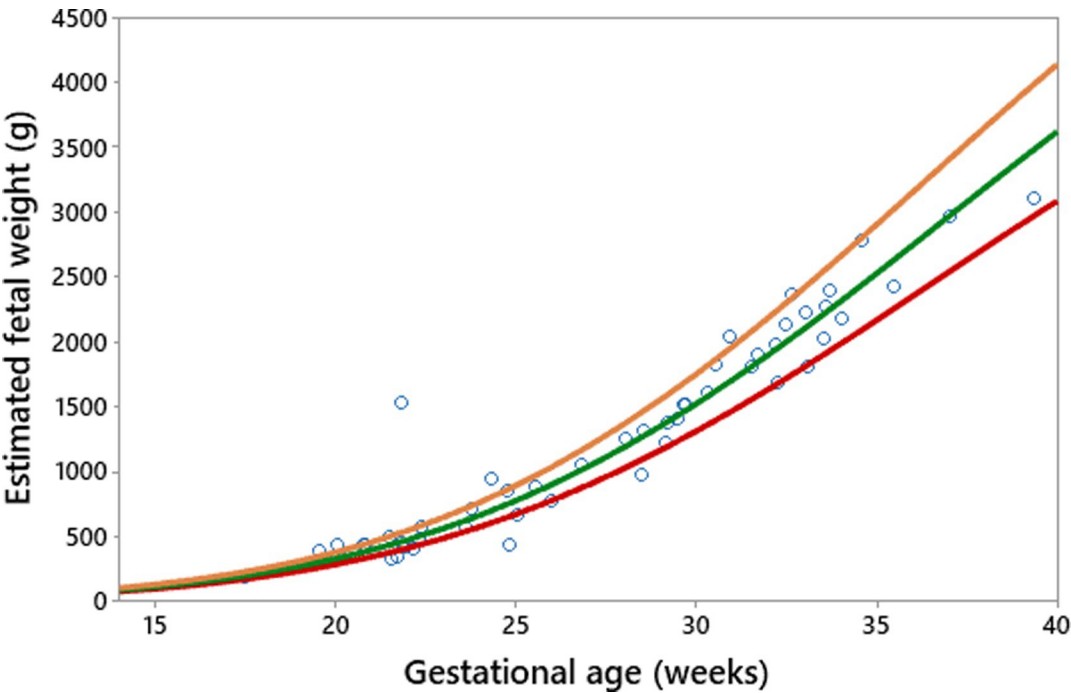

**Fig 4. WHO fetal growth chart applied to an Indonesian population.**

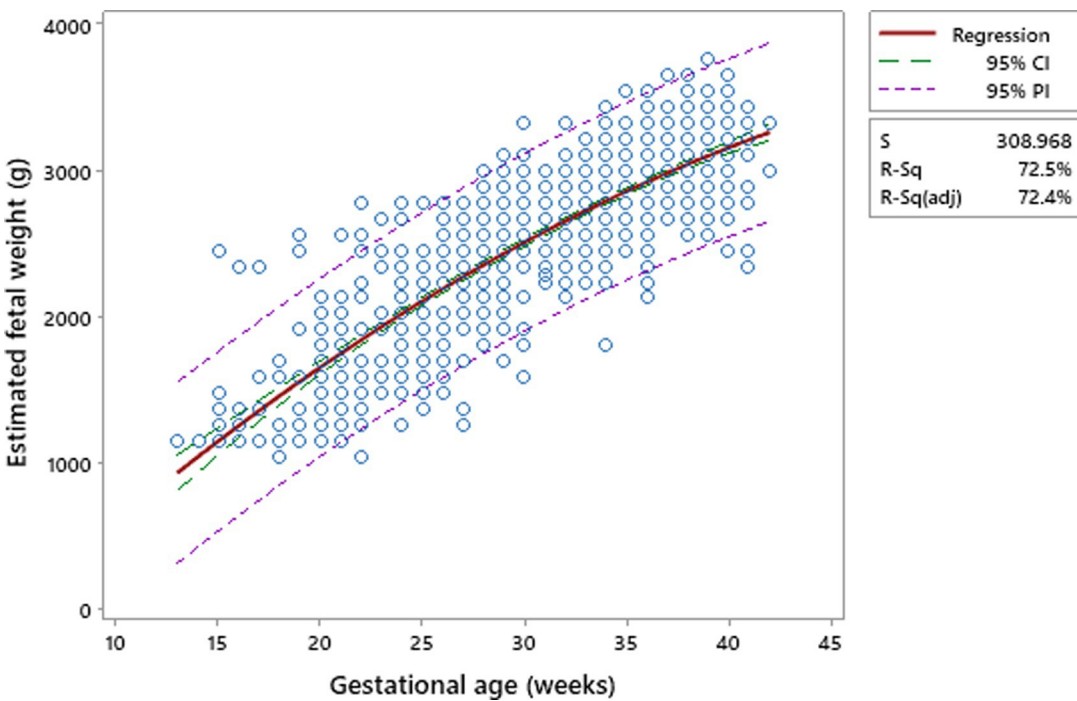

**Fig 5. Scatter plot for the proposed quadratic model.**

et al. [23] was also evaluated using the testing dataset. The results are presented in two periods: during pregnancy (13–36 weeks) (Table 3 and Fig 6) and before or at delivery (32–42 weeks) (Table 4 and Fig 7).

The prediction ability of the proposed clinical-based EFW-GA model and the two existing ultrasound-based EFW-GA models show that the MEs recorded for the proposed model is significantly smaller [731.72 g (between 13 and 36 weeks) and 154.42 g (between 32 and 42 weeks)] than those recorded for the existing models. Similarly, the MPEs, MAPEs, and MEDAPEs recorded for the proposed model are significantly smaller [22.58%, 23.62%, and 21.93%, respectively (between 13 and 36 weeks) and 3.78%, 8.86%, and 8.49%, respectively (between 32 and 42 weeks)] than those recorded for the existing models [> 50%, respectively (between 13 and 36 weeks) and -5.07–4.03%, 9.63–11.92%, and 7.9–10.79%, respectively (between 32 and 42 weeks)].

Tables 3 and 4 summarize that between 13 and 36 weeks, the proposed model produced more predicted values that fall within the 10% and 20% of ABWs (20.1% and 44.3%,

**Table 3. Accuracy of the proposed and existing EFW-GA models (13–36 weeks).**

| | | | Number of pregnant women = 234 with 309 FH observations | | | | |
|---|---|---|---|---|---|---|---|
| Prediction error (ABW–EFW-GA) | Mean (ME) (g) | Mean percentage (MPE) (%) | Mean absolute percentage (MAPE) (%) | Median percentage (MEDPE) (%) | Median absolute percentage (MEDAPE) (%) | Number of estimates within 10% of ABWs (%) | Number of estimates within 20% of ABWs (%) |
| **Proposed models** | | | | | | | |
| Quadratic | 731.72 | 22.58 | 23.62 | 21.93 | 21.93 | 20.06 | 44.34 |
| **Existing models** | | | | | | | |
| Hadlock (1991) | 1635.93 | 51.92 | 52.33 | 55.45 | 55.45 | 7.12 | 15.86 |
| Sotiriadis (2018) | 1684.43 | 53.50 | 53.68 | 56.13 | 56.13 | 4.85 | 13.92 |

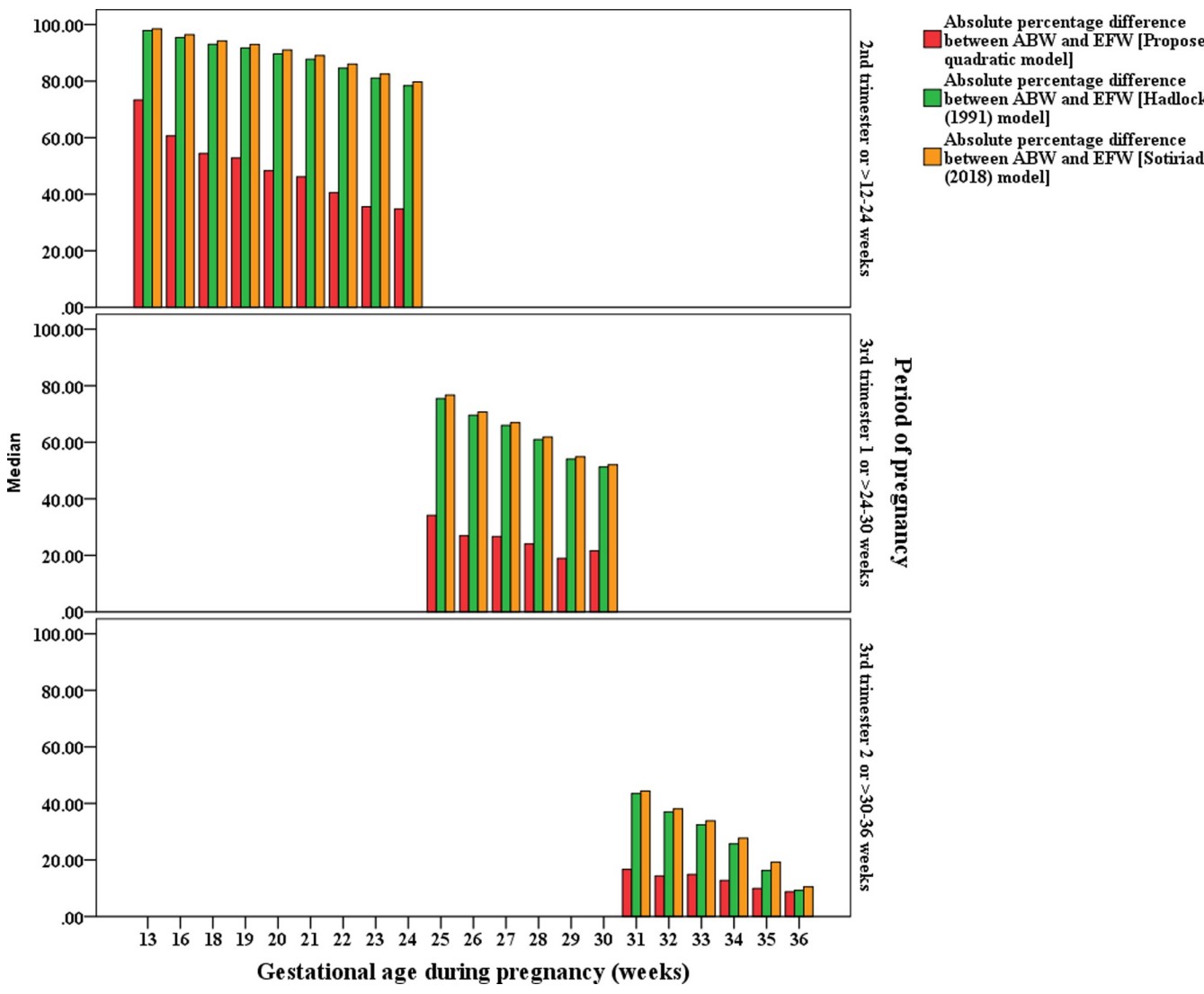

**Fig 6. MEDAPEs of the proposed and existing EFW-GA models (13–36 weeks).**

respectively) compared with the Hadlock model (7.1% and 15.9%, respectively) and the Sotiriadis model (4.9% and 13.9%, respectively). The proposed model was also produced more predicted values that fall within 10% and 20% of ABWs (61.8% and 93.6%, respectively)

**Table 4. Accuracy of the proposed and existing EFW-GA models (32–42 weeks).**

| | | | | | | | |
|---|---|---|---|---|---|---|---|
| Number of pregnant women = 102 with 110 FH observations | | | | | | | |
| Prediction error (ABW–EFW-GA) | Mean (ME) (g) | Mean percentage (MPE) (%) | Mean absolute percentage (MAPE) (%) | Median percentage (MEDPE) (%) | Median absolute percentage (MEDAPE) (%) | Number of estimates within 10% of ABWs (%) | Number of estimates within 20% of ABWs (%) |
| **Proposed models** | | | | | | | |
| Quadratic | 154.42 | 3.78 | 8.86 | 4.07 | 8.49 | 61.82 | 93.64 |
| **Existing models** | | | | | | | |
| Hadlock (1991) | -130.83 | -5.07 | 11.92 | -6.45 | 10.79 | 47.27 | 82.73 |
| Sotiriadis (2018) | 161.47 | 4.03 | 9.63 | 3.96 | 7.90 | 59.09 | 92.73 |

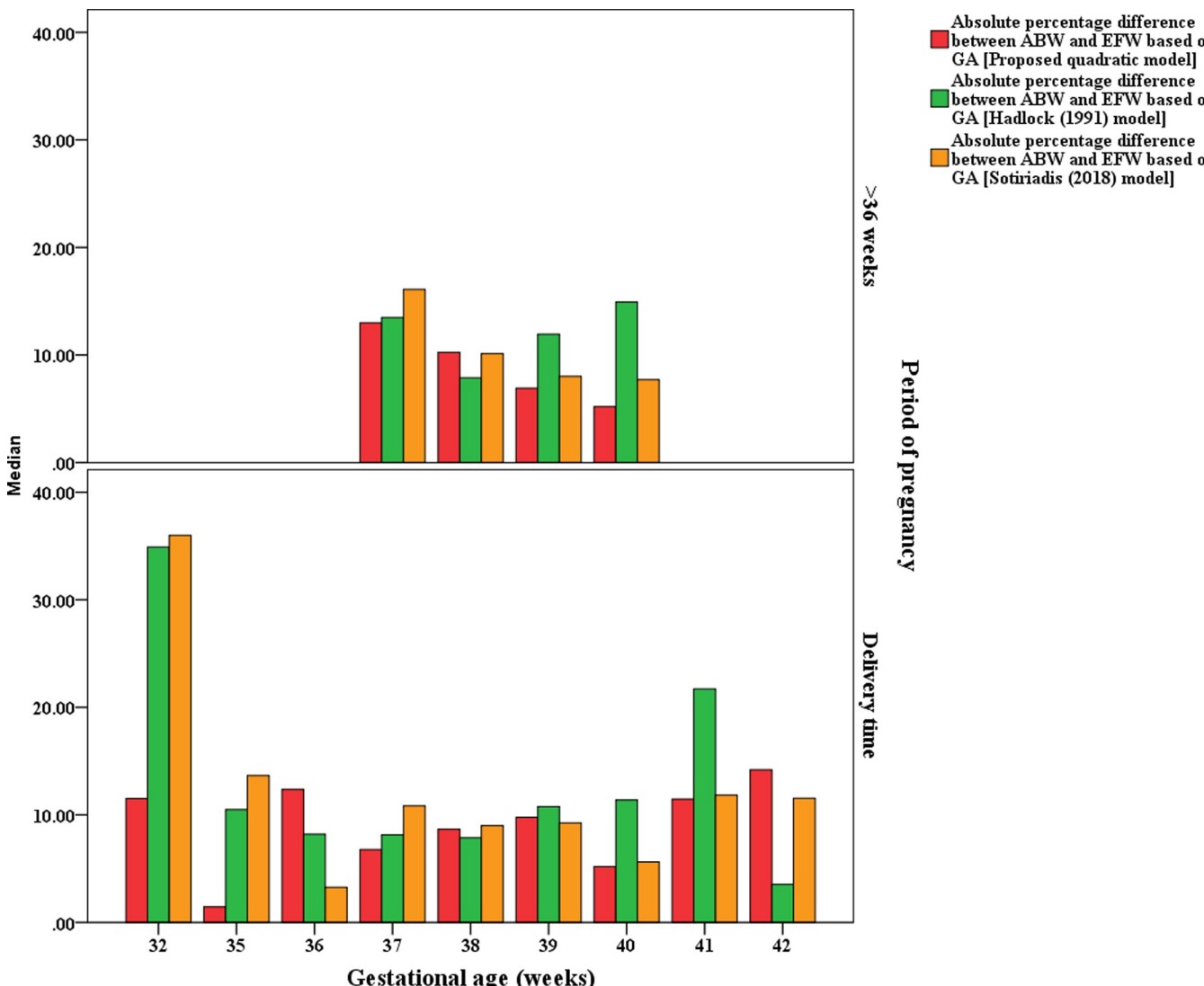

**Fig 7. MEDAPEs of the proposed and existing EFW-GA models (32–42 weeks).**

compared with the Hadlock model (47.3% and 82.7%, respectively) and the Sotiriadis model (59.1% and 92.7%, respectively) when the estimates were made before or at delivery (32–42 weeks). The visualization of these multiple comparisons can be seen in Figs 6 and 7.

The results also show that, between 32 and 42 weeks, there is a significant difference in variances and means of prediction errors between the proposed model and the existing ultrasound models (p-value < 0.05) S8 Table. However, there is no significant difference in the means of prediction errors within the existing models.

**Construction of growth charts for estimated fetal weight based on the proposed EFW-GA quadratic model.** Using the proposed EFW-GA quadratic model based on modified methods of Mikolajczyk et al. [24], the predicted values for specific fetal weight percentiles based on non-ultrasound measurements are listed in Table 5 for each GA. The table shows the percentiles (g) of EFW for an Indonesian population in South Kalimantan province with a mean birth weight at 40 weeks of gestation of 3178.4 g.

**Table 5. Estimated fetal weights for an Indonesian population.**

| Gestational age (weeks) | Percentiles of estimated fetal weight (g) | | | | | | | | | | |
|---|---|---|---|---|---|---|---|---|---|---|---|
| | 1st | 3rd | 5th | 10th | 25th | 50th (mean) | 75th | 90th | 95th | 97th | 99th |
| 20 | 1271 | 1346 | 1386 | 1448 | 1551 | 1666 | 1781 | 1884 | 1946 | 1986 | 2062 |
| 21 | 1343 | 1423 | 1466 | 1531 | 1640 | 1762 | 1883 | 1992 | 2057 | 2100 | 2180 |
| 22 | 1415 | 1499 | 1544 | 1612 | 1727 | 1855 | 1983 | 2097 | 2166 | 2211 | 2295 |
| 23 | 1484 | 1573 | 1619 | 1692 | 1812 | 1946 | 2080 | 2201 | 2273 | 2320 | 2408 |
| 24 | 1552 | 1645 | 1694 | 1769 | 1895 | 2035 | 2175 | 2301 | 2377 | 2426 | 2518 |
| 25 | 1619 | 1715 | 1766 | 1845 | 1976 | 2122 | 2268 | 2400 | 2479 | 2530 | 2626 |
| 26 | 1683 | 1784 | 1837 | 1919 | 2055 | 2207 | 2359 | 2496 | 2578 | 2631 | 2731 |
| 27 | 1747 | 1851 | 1906 | 1991 | 2133 | 2290 | 2448 | 2590 | 2675 | 2730 | 2834 |
| 28 | 1808 | 1916 | 1973 | 2061 | 2208 | 2371 | 2534 | 2681 | 2769 | 2826 | 2934 |
| 29 | 1868 | 1980 | 2039 | 2129 | 2281 | 2450 | 2618 | 2770 | 2861 | 2920 | 3031 |
| 30 | 1927 | 2042 | 2102 | 2196 | 2353 | 2526 | 2700 | 2857 | 2951 | 3011 | 3126 |
| 31 | 1984 | 2102 | 2165 | 2261 | 2422 | 2601 | 2780 | 2941 | 3038 | 3100 | 3218 |
| 32 | 2039 | 2160 | 2225 | 2324 | 2490 | 2674 | 2858 | 3023 | 3122 | 3187 | 3308 |
| 33 | 2093 | 2217 | 2283 | 2385 | 2555 | 2744 | 2933 | 3103 | 3204 | 3271 | 3395 |
| 34 | 2145 | 2273 | 2340 | 2445 | 2619 | 2812 | 3006 | 3180 | 3284 | 3352 | 3480 |
| 35 | 2195 | 2326 | 2395 | 2502 | 2680 | 2878 | 3077 | 3255 | 3362 | 3431 | 3562 |
| 36 | 2244 | 2378 | 2449 | 2558 | 2740 | 2943 | 3145 | 3327 | 3436 | 3507 | 3641 |
| 37 | 2291 | 2428 | 2500 | 2612 | 2798 | 3005 | 3211 | 3398 | 3509 | 3581 | 3718 |
| 38 | 2337 | 2477 | 2550 | 2664 | 2854 | 3065 | 3276 | 3465 | 3579 | 3653 | 3792 |
| 39 | 2381 | 2523 | 2598 | 2714 | 2908 | 3123 | 3337 | 3531 | 3647 | 3722 | 3864 |
| 40 | 2424 | 2568 | 2645 | 2763 | 2960 | 3178 | 3397 | 3594 | 3712 | 3788 | 3933 |
| 41 | 2465 | 2612 | 2690 | 2809 | 3010 | 3232 | 3455 | 3655 | 3775 | 3852 | 3999 |

Fig 8 demonstrates the distribution of actual (raw) observations of predicted values based on the EFW-FH model (using testing data, n = 282 with 419 observations) superimposed on the EFW curve developed based on the proposed quadratic EFW-GA model together with the 95% confidence interval. The distribution of the actual observations shown in Fig 8 was then superimposed on the existing profile limits proposed by Mikolajczyk, et al. [24], the Intergrowth 21st project reference (International standard) [19] and the WHO [5]. The result is shown in Fig 9.

It can be seen from Fig 9 that the existing fetal growth charts for EFW do not fit well to the local population because they underestimate. The patterns of fetal growth measured by clinical measurements are different to the ones measured by ultrasound.

Furthermore, fetal birth weight at term pregnancy, between 37 and 41 weeks, was compared using the 50th percentile predicted values of fetal weight from the developed quadratic model and the ABW of the study population. These results were also compared with the predicted values of fetal weight, and the observed values of birth weight derived from the existing models [22, 23] and references [5, 19]. This comparison is presented in Table 6.

It can be seen from Table 6 that generally the mean of predicted fetal weights [based on the proposed model, Equation (2)] in the Indonesian population was lower than those predicted based on the existing models and references, i.e., those developed based on ultrasound measurements of fetal biometrics. At 38 weeks, the expected value was comparable with the one derived from the existing model [23] and the existing reference from the Intergrowth 21st Project [19]. Meanwhile, the mean of actual birth weights in the study population was also lower than the one observed in the existing model [22].

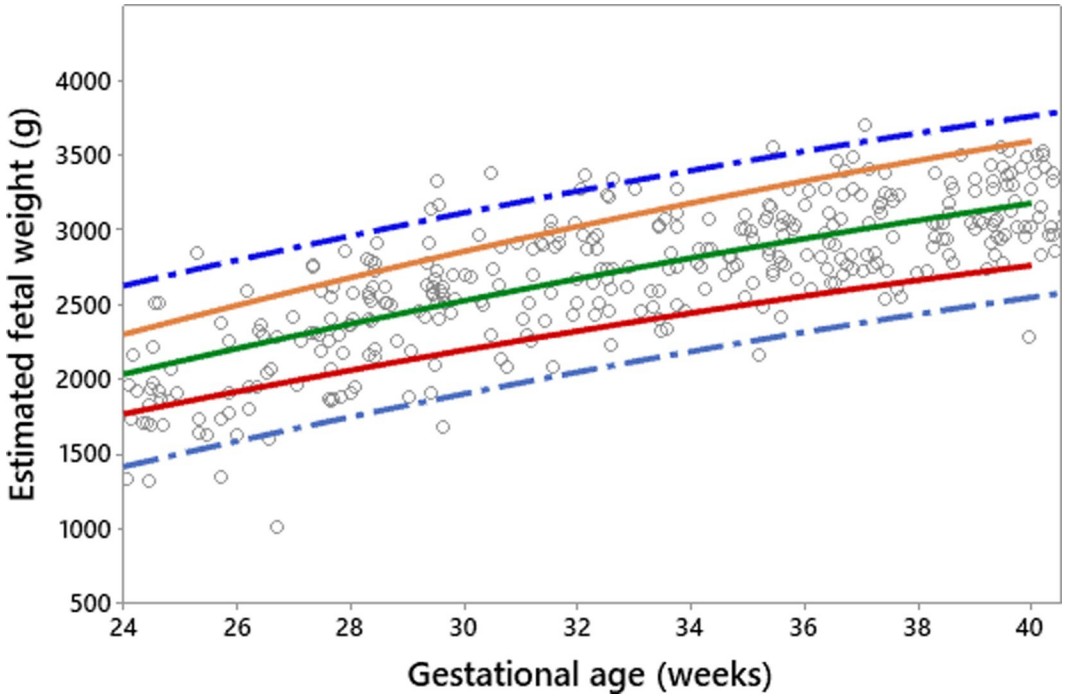

**Fig 8. Proposed fetal weight chart with test data superimposed.** Key: ---- = 10th percentile, ---- = 50th percentile, ---- = 90th percentile, ---- = 95% confidence intervals.

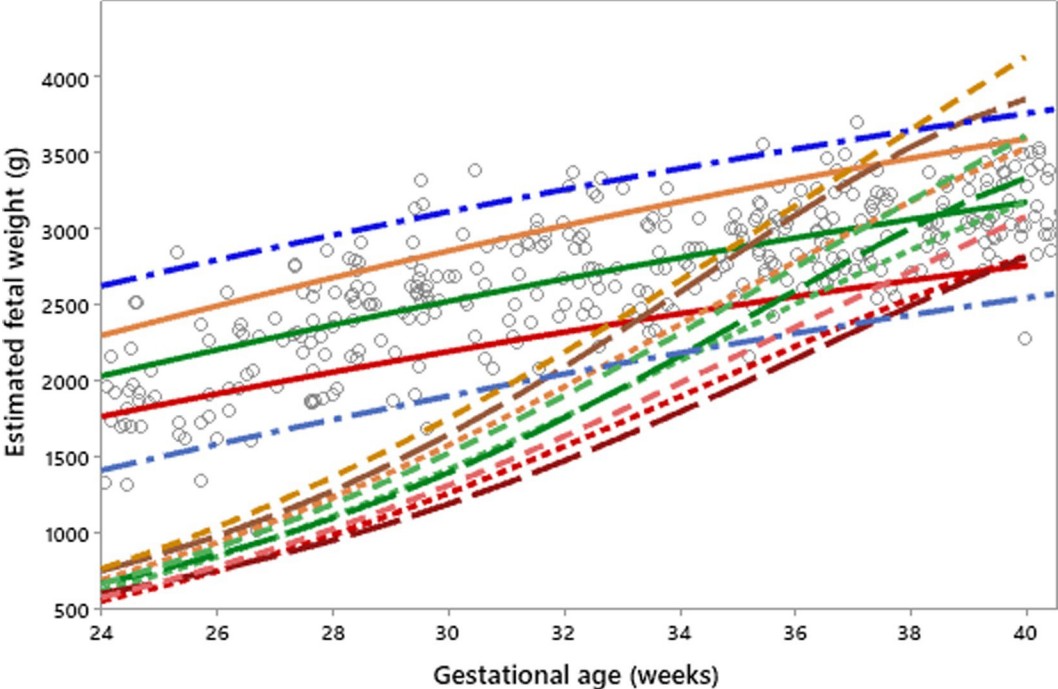

**Fig 9. Fit of proposed weight chart compared with existing references.** Key: as for Fig 8 with profile limits added from -------- = global reference [24] — — = Intergrowth 21st Project reference (International standard) [19] – – – – = WHO [5].

**Table 6. Fetal birth weight at term pregnancy (37–41 weeks).**

| Gestational age (week) | Weights (g) | | | | | | |
|---|---|---|---|---|---|---|---|
| | Present study (predicted values*) | Present study (observed values^) | Hadlock (1991) (predicted values*) | Hadlock (1991) (observed values^) | Sotiriadis (2018) (predicted values*) | Stirnemann (2017) (predicted values*) | Kiserud (2017) (predicted values*) |
| 37 | 3005 | 3027 (10) | 3028 | NA | 2769 | 2806 | 2966 |
| 38 | 3065 | 3127 (9) | 3236 | 3234 | 2992 | 3006 | 3186 |
| 39 | 3123 | 3142 (11) | 3435 | 3469 | 3213 | 3186 | 3403 |
| 40 | 3178 | 3178 (10) | 3619 | 3598 | 3426 | 3338 | 3617 |
| 41 | NA | 3112 (12) | NA | 3686 | NA | NA | NA |

Note:

Numbers in parentheses are standard deviations expressed as a percentage of the actual mean birth weight

NA = No data or insufficient data at these time points

* The 50th percentile weight for GA based on the model

^ The mean of actual birth weights

Fig 10 represents the distribution of actual (raw) data of predicted values for the 16 LBW newborns among the 435 singleton live newborns in the prospective data, between 20 and 40 weeks, using the developed fetal growth chart.

The capability of the developed clinical-based fetal growth chart to detect abnormal patterns among 282 normal (Fig 8) and 16 LBW newborns (Fig 10) was assessed by calculating

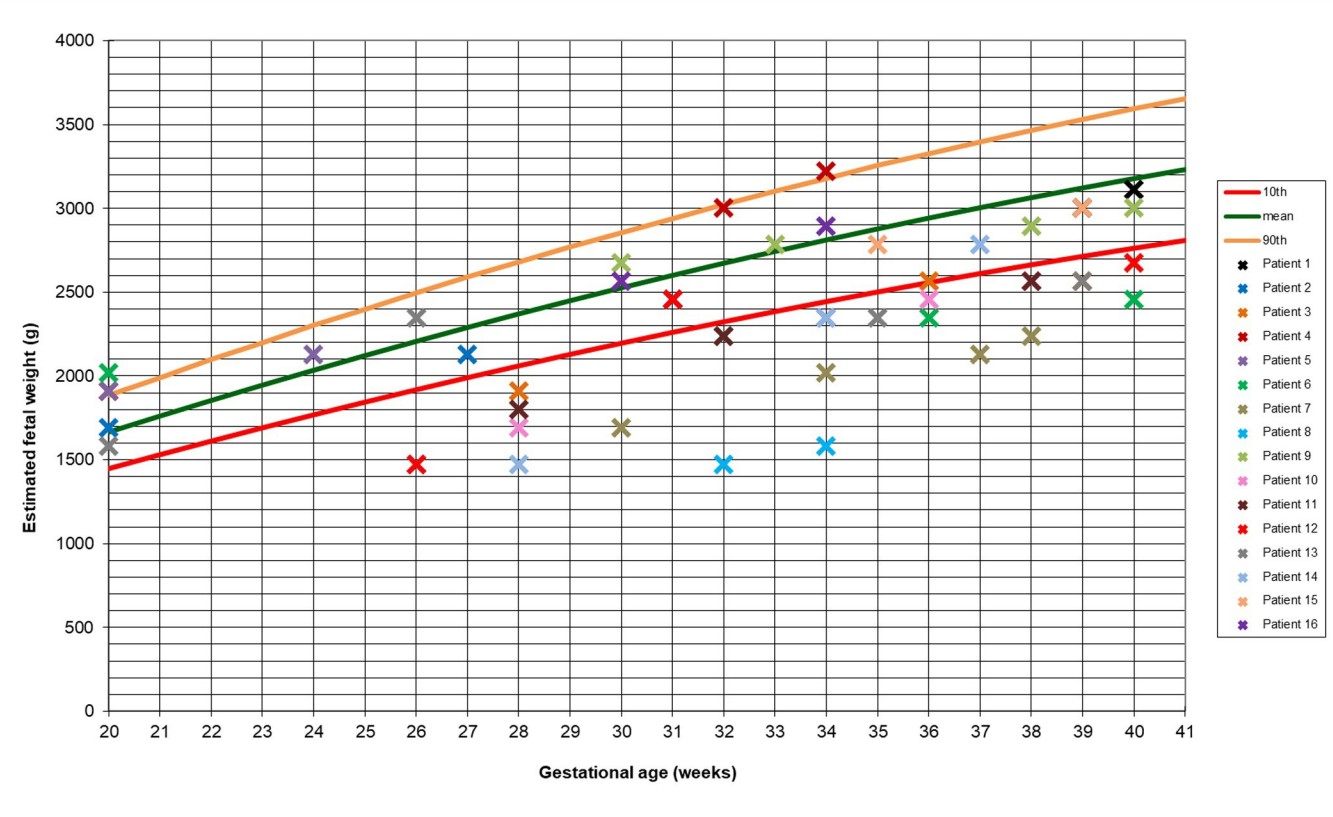

**Fig 10. Proposed fetal weight chart (20–40 weeks) used with data relating to 16 LBW babies.**

the percentage of cases falling below the LSL (the 10th percentile), within the 10th and the 90th percentiles and above the USL (the 90th percentile). The results are shown in Table 7.

Based on the data in Table 7, the developed fetal growth chart was able to detect 43.8–68.8% of the abnormal patterns of EFW during pregnancy among pregnant women who delivered LBW babies. When fetuses were delivered with normal weight, the risk of having such unusual growth patterns was smaller (20.9–54.3%). It shows that for 16 LBW babies, 7 (43.8%) had at least one EFW reading falling below the proposed 10th percentile and 11 (68.8%) at least one EFW reading below the 50th percentile.

## Discussion

### Review of existing growth charts for estimated fetal weight and evaluation of their applicability in low resource settings

Fetal growth charts have not been introduced in the Indonesian first level of health care systems. The growth charts are also not included in the current MCH booklet (Buku KIA) that is used to monitor individual mothers, fetuses, and newborns during ANC and postnatal care [12, 15, 16]. This study, therefore, set out with the aim of filling this gap, first by reviewing the existing studies on fetal growth charts, particularly for EFW. Potential challenges when implementing such surveillance tools in Indonesian primary health care centers were also statistically reviewed. This local review is necessary to ensure the fitness and feasibility of the charts to the Indonesian population before their implementation [5, 8].

Based on the literature review, the selected studies on fetal growth charts have used previously published ultrasound-based statistical models to estimate fetal weight. However, in the majority of limited resource settings, the provision of ultrasound machines and skilled personnel is logistically not always available [1, 29]. It has already been noted that the ultrasound method to estimate the fetal weight is not universally accessible in the current practice of ANC across Indonesian primary health care centers [12, 15]. This hinders fetal weight estimation using the ultrasound prediction models that are commonly used in the existing fetal growth chart studies.

A fetal weight prediction model based on the clinical measurement of FH has been proposed in the previous study [30]. This model was used in the development of one of the existing customized fetal growth charts as an alternative in the absence of ultrasound facilities. The model was constructed based on a linear relationship between EFW and FH, which was measured simultaneously during the third trimester of pregnancy. However, the EFW was derived from the extrapolation technique of Campbell and Wilkin [17] and the formulas of Hadlock,

**Table 7. Signals of fetal growth abnormality.**

| Raw data (g) | Case | Gestational age (weeks) | Measurement | N | Within the optimal fetal growth limits | Outside the optimal fetal growth limits | | |
|---|---|---|---|---|---|---|---|---|
| | | | | | Between the 10th and 90th percentiles | Below the 10th percentile | Below the 50th percentile | Above the 90th percentile |
| EFW Model 5 (The optimal clinical-based prediction model based on FH) | Normal newborns | Between 13 and 42 weeks | Individual pregnant women | 282 | **186 (66.0%)** | 59 (20.9%) | 153 (54.3%) | 37 (13.1%) |
| | | | Repeated observations | 419 | **302 (72.1%)** | 70 (16.7%) | 228 (54.4%) | 47 (11.2%) |
| | LBW newborns | Between 13 and 39 weeks | Individual pregnant women | 16 | 6 (37.5%) | **7 (43.8%)** | **11 (68.8%)** | 3 (18.8%) |
| | | | Repeated observations | 40 | 18 (45.0%) | **18 (45.0%)** | **29 (72.5%)** | 4 (10.0%) |

et al. [18] and Mongelli, et al. [31]. Therefore the clinical-based prediction model proposed in their study still depends on ultrasound data.

A minimum dataset of individual maternal, fetal, and neonatal characteristics is required to utilize the existing customized fetal growth charts. This information is used to derive the regression coefficients for adjusting the term optimal weight [27]. However, access to this information remains challenging in most developing countries, particularly in the majority of Indonesian primary health care centers, due to low quality in the routine collection of ANC data [12, 15].

As already mentioned in our previous publication [15], the average amount of recorded ANC data across primary health care providers in the South Kalimantan province, between 2012 and 2016, was approximately 17.5%. This result is lower than the national reported figure of 42.5% [32]. For this reason, the existing customized fetal growth charts are less applicable in Indonesia.

## Local evaluation of prediction models used in the development of existing growth charts for estimated fetal weight and the comparison with the proposed model

This study has assessed the accuracy of the ultrasound-based prediction models used in the development of the existing fetal growth charts and compared them with the proposed clinical model before implementation in the Indonesian population. Such validation is highly recommended to ensure the fitness and practicability of charts for the local population [5, 8].

Based on the comparison analysis, the proposed model produced fewer prediction errors than the existing models both during pregnancy (between 16 and 38 weeks) and before or at delivery (between 33 and 40 weeks). The recorded MPE was 7.1% in those born within an average of 14 days (2 weeks between the last scan and birth) (n = 19). This error is smaller than in the previous research (-10.7%) on babies born precisely 14 days after the last scan (n = 196). In the previous publication [6], it has been shown that a weight prediction model based on FH only produced fewer errors compared with the existing clinical and ultrasound models. Therefore, the proposed model, as the improvement of the previously published model [6], can be a potential weight prediction model to be used in the development of an alternative fetal growth chart for EFW. This is particularly significant for those who are living in rural areas where ultrasound is not always accessible.

Ultrasound measurement, as a complement to the routine measurement of FH, is necessary only if it is considered appropriate. Regular analysis of FH is recommended for low-risk pregnancies to screen the intrauterine development of the fetus [1, 3, 33]. On the other hand, ultrasound measurement of fetal characteristics is recommended only for high-risk pregnancies to monitor the growth of the fetus [1, 3].

Two standard fetal growth charts for EFW, one proposed by the Intergrowth 21st Project and one by WHO, were evaluated for their fit to the Indonesian population [5, 19]. Based on the analysis, both charts could potentially be implemented in Indonesian primary health care centers to monitor fetal growth provided that information on fetal biometric characteristics (required to estimate fetal weight) is available. The former chart uses its own EFW based on AC and HC, while the latter chart uses the EFW model previously published by Hadlock et al. [18], which is based on AC, HC, and FL. However, their practicality remains challenging for Indonesian ANC settings due to the lack of recorded data on fetal biometric characteristics during pregnancy [6, 12, 15]. Therefore, an alternative weight prediction model using clinical measurement (rather than ultrasound measurement) is required to develop a suitable fetal growth chart for EFW. The chart can be easily implemented in the local primary health care centers or other settings with low health resources.

## The development of an alternative growth chart for estimated fetal weight in the absence of ultrasound

The proposed quadratic clinical model was selected as the most suitable formula to be used to develop an alternative fetal growth chart for EFW in settings with limited resources (no ultrasound). The proposed quadratic regression model has produced more effective estimates of fetal weight than the existing ultrasound formulas. Furthermore, the proposed formula for estimating fetal weight is simple and can be used by doctors, midwives, and pregnant women. This would improve the quality of communication, information, and education as part of routine ANC service in low-resource primary health care centers.

The quadratic model fitted the local data more efficiently compared to the other estimating models, particularly between 24 and 40 weeks of gestation. This finding seems to be consistent with other research, which found that the quadratic model was the optimal prediction model for fetal weight between the second and third trimesters of pregnancy [22, 23, 34]. Therefore, in this study, the quadratic model has been used to develop an alternative fetal growth chart.

The prediction accuracy of the optimal model, i.e., the quadratic model, has also been assessed using a prospective dataset and compared with the existing regression models based on ultrasound measurements [22, 23]. The MPE for the proposed model was 22.6% in those born between 13 and 36 weeks (prospective; n = 234). This percentage error was lower than the existing ultrasound-based prediction models [22, 23] when implemented on local data (MPEs 51.9–53.5%). Using data recorded between 32 and 42 weeks (before or at delivery), the results showed that the proposed quadratic model produced smaller ME (154.4g), less MPE (3.8%), and less MAPE (8.9%). The MPE steadily tended towards zero as the time interval between the last scan and birth decreased [19]. The MAPE was higher than that reported by Hadlock et al. [22] when they applied their model in their original population (1.1%) but smaller than the one indicated in the existing study when Hadlock et al. 's model was used to the local population (11.9%).

The results show that the difference between EFW during pregnancy (between 13 and 36 weeks) and ABW was more significant than when it was measured before delivery (between 32 and 42 weeks). This finding is in agreement with Hadlock's earlier study of fetal growth [22] and also the most recent study reported in the literature [23]. The divergence between predicted and observed term birth weights reported in this study (MPE = 3.8% and MAPE = 8.9%) are higher than those published by Hadlock (MPE = 0.8% and MAPE = 1.1%), while the average variability reported in this study (when expressed as a percentage of the predicted values) is smaller (11%) (Table 5) than in the earlier study (13%) [22]. Also, at earlier GAs, for example, at 27 weeks, the difference between ABWs and EFWs reported in this study was smaller (11%) than that published in Sotiriadis' recent study (14%) [23].

There is no fetal growth chart for EFW available in the current ANC practice of Indonesian primary health care centers. Its absence is particularly noticeable in the MCH booklet (Buku KIA) [16], which is nationally recognized as one of the vital profile monitoring records during pregnancy. One of the reasons for this is the difficulty in collecting the data required to develop a standard fetal growth chart [11, 12, 15]. This study, therefore, set out with the aim of creating a fetal growth chart for use in the primary health care settings in Indonesia.

Customized and standard fetal growth charts for EFW have been developed and are highly recommended for general use when local data are not available. The charts can be used to monitor fetal growth, defined as the change in EFW with GA. If access to ultrasound measurement of fetal biometric characteristics is limited, this hinders fetal weight estimation using the existing models. Low-quality data (missing or incomplete records) on the minimum data requirements on individual maternal, fetal, and neonatal characteristics also make the existing

customized charts less applicable in the local setting. These have been identified statistically as some of the potential challenges in implementing the existing charts in the Indonesian primary health care centers.

The fetal growth charts are developed to monitor the growth patterns at different GAs and ensure that they follow normal predicted values and limits for fetal weight so that the optimal birth weight and a safe pregnancy outcome can be achieved. Since GA is not a significant predictor for birth weight [19], a prediction model of fetal weight is commonly developed based on FH or ultrasonic measurements. This EFW model was then regressed by GA to generate a growth chart for EFW. As a result, most of the existing fetal growth charts for EFW have been developed based on regression models between EFW and GA, i.e., derived based on ultrasound-based prediction models using fetal biometric characteristics [5, 19, 24, 26, 27].

However, the use of ultrasound-based prediction models to estimate fetal weight makes it challenging to implement these existing charts for the Indonesian population due to a lack of access to actual fetal biometric data. This study proposed an alternative fetal growth chart based on the most effective clinical-based EFW-FH model (based on FH measurement) for predicting fetal weight at a given GA. This chart can be easily implemented in settings with limited resources using readily accessible clinical information rather than ultrasound data.

A wide range of statistical methods has been implemented to construct GA-related charts for pregnancy dating, fetal size, and fetal weight [35], mainly using a parametric approach from cross-sectional to longitudinal data. The aim of this study, however, is to create a fetal growth chart for use in Indonesia to make up for the absence of fetal growth charts in settings with limited resources. Therefore simpler yet still appropriate techniques would preferably be used to develop this chart.

This study, for the first time, has developed an alternative fetal growth chart for EFW based on a weight prediction model that uses easily accessible, statistically significant Indonesian clinical data, the maternal FH. It is essential to highlight the significance of the developed growth chart for risk assessment during ANC to prevent LBW delivery. Using prospective data, it has been shown that the proposed chart can effectively detect signs of abnormality with the risk of LBW delivery between 20 and 41 weeks. The introduction of the chart can assist midwives and other medical practitioners to identify high-risk pregnancies, prevent delays in making decisions, referrals, and interventions, and reduce the number of unnecessary investigations during ANC. The utilization of this chart in primary health care centers, as the first level of healthcare in Indonesia and located near where people live, should be prioritized.

The development of the fetal growth chart followed the previously published procedures of Mikolajczyk et al. [24]. Their methods combine the fetal weight estimation with the notion of proportionality, as proposed by Gardosi, et al. [26], and adjust it to the mean birth weight at 40 weeks of gestation for any local population. However, the current methods used one of the ultrasound-based prediction models, the one proposed by Hadlock, et al. [18], which was based on fetal measurements of biparietal diameter (BPD), HC, AC and FL to estimate fetal weight. Then the model was regressed by GA [22] to construct the fetal growth chart.

It has also been suggested that the existing reference charts, i.e., those developed based on EFW and GA, should not be applied for monitoring fetuses whose ages (GA) were not verified using ultrasound [22]. Therefore a quadratic model was developed based on regression analysis between EFW (based on FH) and GA, and this was used to create the alternative fetal growth chart.

The mean birth weight at 40 weeks in the local population was calculated from 142 pregnant women who delivered live singleton newborns with normal weight. Following the recommendations of Mikolajczyk, et al. [24], the size of this sample population at 40 weeks agrees with the population sample size recommended providing acceptable accuracy, i.e., a minimum

of 100 samples per GA, and with the inclusion criteria for reference birth weights of a population. The local SD is smaller (10.1%) than the empirical SD in most countries that participated in 2004–2008 WHO Global Survey on Maternal and Perinatal Health (average 13.2%, i.e., 13.2% of the mean weight at 40 weeks) [24]. The result implies that the local population was more homogeneous in birth weight distribution at 40 weeks than other countries involved in the WHO global survey.

Since the aim is to reduce the prevalence of LBW newborns, this study concentrated only on the detection of EFW below the 10th and 50th percentiles. It is important to highlight the significance of the proposed EFW growth chart for risk assessment during ANC to prevent LBW newborns, and to repeat that there is currently no equivalent in use in Indonesia. Evaluation of the data from the 16 women who had LBW babies out of the 435 pregnant women studied shows that the proposed chart can effectively detect signs of abnormality.

A predominant strength of this research was its methods and data sources. This research has implemented a prospective study using more reliable information, which is the ideal method to monitor and record changes in the process of pregnancy from the start to delivery. This longitudinal study approach has also been recognized as the most recommended design for studies in fetal growth [36, 37].

This study also encountered limitations associated with the accuracy of the information recorded on the manual pregnancy register or with inaccurate data transferred to the electronic database. This limitation was minimized by monitoring and controlling the process of data transfer to reduce potential error.

## Conclusions

The proposed clinical model has a comparable ability in predicting fetal weight with less error than the existing models and is, in fact, even more, effective at earlier GAs than the existing models. Consequently, the proposed model could be an alternative model to estimate fetal weight and develop a suitable fetal growth chart. The presence of this alternative chart would be particularly significant for those who are living in rural areas where ultrasound facilities are not always available. This would improve the quality of fetal risk assessment during pregnancy to detect fetal growth abnormalities and reduce the risk of adverse neonatal outcomes.

In conclusion, the outcomes of this research can provide useful administrative and scientific guidelines for the expansion of health services and programs and the effective distribution of limited government resources in rural areas. This includes analysis of where further aid investments are likely to the best impact on reducing neonatal mortality. The outcomes can also effectively aid midwives and other medical practitioners in identifying the key risk factors and types of clinical interventions required before delivery to reduce the mortality rate.

## Supporting information

**S1 Fig. Recruitment of participants.**
(PDF)

**S2 Fig. Recruitment of study participants.**
(PDF)

**S1 Table. Selected studies on fetal growth charts for estimated fetal weight.**
(PDF)

**S2 Table. Baseline characteristics and antenatal events.**
(PDF)

**S3 Table. Two-sample F-test and T-test results (16–38 weeks).**
(PDF)

**S4 Table. Two-sample F-test and T-test results (33–40 weeks).**
(PDF)

**S5 Table. Baseline characteristics and antenatal events.**
(PDF)

**S6 Table. Regression models between EFW and GA.**
(PDF)

**S7 Table. Analysis of residuals.**
(PDF)

**S8 Table. Two-sample F-test and T-test results (32–42 weeks).**
(PDF)

## Acknowledgments

The authors are incredibly thankful to the head of the provincial health department of South Kalimantan, Dr. H. Achmad Rudiansyah, M.Sc., who supported and permitted time release for the representative midwives to attend the training. We also thank Nani Lidya, SKM for her participation in the training as the representative of the provincial health department of South Kalimantan, Dr. Bambang Abimanyu, Sp.OG., KFM for his contribution in providing information regarding fetomaternal medicine, and Dr. Andy Yussianto, M.Epid for his role in providing information about current maternal and child health surveillance in Indonesia.

We are immensely appreciative of the head of the provincial partner midwife Association (IBI) of South Kalimantan, Hj. Tut Barkinah, S.Si.T., M.Pd., and her members, Hj. Nurtjahaya, S.ST., Hj. Masjudah, S.ST., and Hj. Supri Nuryani, S.Si.T., M.Kes., who supported and selected the representative midwives to participate in the training.

The authors are greatly indebted to the HDR Language and Learning Advisor of RMIT University, Dr. Judy Maxwell, for her role in providing language help and proofreading the article.

We are greatly appreciative of the midwives team[*] in their role in gathering the prospective data (June 1, 2016—June 30, 2017) from their assigned workplace.

[*]Midwives team

Hj Ariati, S.ST (Banjarmasin), Sari Milayanti, AMd.Keb and Hj. Masjudah, S.ST (Banjarbaru), Rini, AMd.Keb (Banjar), Rahmi Widiati, AM.Keb (Barito Kuala), Suwarni, AM.Keb (Tapin), Hj Hiriana, AMd.Keb (Hulu Sungai Selatan), Yanti Pertiwi, AMd. Keb (Hulu Sungai Tengah), Hj. Siska Yunita, AM.Keb (Hulu Sungai Utara), Nurjanah, S.Si.T (Balangan), Suparti, S.Si.T (Tabalog), Rina, AM.Keb (Tanah Laut), Sri Wahyuningsih, AMd.Keb (Tanah Bumbu), Yani Kristanti, AMd. Keb (Kotabaru), Hj Raihatul Jannah, AMd. Keb (BPM Banjarmasin), Rinawati, AMd. Keb (BPM Banjarmasin), Eka Septina, AMd. Keb (BPM Banjarmasin), Noorjannah, S.ST (BPM Banjarmasin), and Fauziah Olfah, S.ST (BPM Kotabaru).

## Author Contributions

**Conceptualization:** Dewi Anggraini, Mali Abdollahian.

**Data curation:** Dewi Anggraini.

**Formal analysis:** Dewi Anggraini, Mali Abdollahian, Kaye Marion.

**Investigation:** Dewi Anggraini.

**Methodology:** Dewi Anggraini, Mali Abdollahian, Kaye Marion.

**Project administration:** Dewi Anggraini.

**Resources:** Dewi Anggraini.

**Software:** Dewi Anggraini.

**Supervision:** Mali Abdollahian, Kaye Marion.

**Validation:** Dewi Anggraini.

**Visualization:** Dewi Anggraini.

**Writing – original draft:** Dewi Anggraini.

**Writing – review & editing:** Mali Abdollahian, Kaye Marion.

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
