## [Decision Letter · Decision Letter 0]

20 Jul 2020

PONE-D-19-33469

The development of an alternative growth chart for estimated fetal weight in the absence of ultrasound: application in Indonesia

PLOS ONE

Dear Dr. Anggraini,

Thank you for submitting your manuscript to PLOS ONE. After careful consideration, we feel that it has merit but does not fully meet PLOS ONE’s publication criteria as it currently stands. Therefore, we invite you to submit a revised version of the manuscript that addresses the points raised during the review process.

We look forward to receiving your revised manuscript.

Kind regards,

Giovanni Delli Carpini

Academic Editor

PLOS ONE

Journal Requirements:

a) Did participants provide their written or verbal informed consent to participate in this study?

Additional Editor Comments (if provided):

Please revise the manuscript according to Author's instructions and Manuscript preparation guidelines.

Reviewers' comments:

Reviewer's Responses to Questions

**Comments to the Author**

1. Is the manuscript technically sound, and do the data support the conclusions?

Reviewer #1: Yes

Reviewer #2: Yes

2. Has the statistical analysis been performed appropriately and rigorously? 

Reviewer #1: I Don't Know

Reviewer #2: Yes

3. Have the authors made all data underlying the findings in their manuscript fully available?

Reviewer #1: Yes

Reviewer #2: Yes

4. Is the manuscript presented in an intelligible fashion and written in standard English?

Reviewer #1: Yes

Reviewer #2: Yes

5. Review Comments to the Author

Reviewer #1: No, i don't have any comments for authors. But he did not follow author's instructions and manuscript preparation guidelines.

Author needs to follow journal's manuscript preparation guidelines in order to create good manuscript.

Reviewer #2: The present study entitled “The development of an alternative growth chart for estimated fetal weight in the absence of ultrasound: application in Indonesia” by Anggraini Dewi at al. showed that the proposed model based on maternal fundal height has a comparable ability in predicting fetal weight with less error than existing model and it would improve the quality of fetal risk assessment during pregnancy and reduce the risk of adverse neonatal outcomes. Despite relevant topic and public health importance of the study, there are many issues related to presentation of results, discussions and conclusions which need to be addressed for strengthening the paper in order to maintain readers’ clarity and interest.

Major revisions:

• In results, while describing the general characteristics of study population, total number of enrolled women is missing and without this, it is difficult to understand various percentages given in this section.

• While describing tables 1 and 2, authors have reported all estimates (ME, MPE, MAPE, PEDPE, etc.) in lines from 285 to 297. This text should be rephrased to simplify the message of both tables.

• In Tables 1, all median percentages and median absolute percentages for various models are same, whereas in table 2, both values are different for all models except Stirnemann (2017). These should be rechecked.

• The lines 320 and 321 regarding F test should be removed from here as it has already been described in lines 134 – 136 in methods section.

• Lines 373 and 374 “However, since ………. Problems.” Should be part of discussion.

• Lines 413 and 414 on page 19 regarding F test should be moved to methods.

• Lines 418 – 420 can be moved to discussion/conclusion. Similarly, lines 457 – 459 on page 21 “The reason ………….EFW-FH model” should be moved to discussion.

• The text “Since the aim ………………percentiles” in lines 501 and 502 should be the part of discussion

• The conclusion part should be revised as most of the text given in 1st and 3rd para of conclusions should be part of discussion. Similarly, the strengths and limitations of the study given on pages 32 and 33 under conclusion section should be moved to discussion section.

Minor corrections:

• The abstract can be presented with standard headings, such as Introduction, methods, results, and Conclusion.

• Some language corrections will further improve the quality of paper.

6. PLOS authors have the option to publish the peer review history of their article (what does this mean?). If published, this will include your full peer review and any attached files.

Reviewer #1: No

Reviewer #2: No

---

## [Author Response · Author response to Decision Letter 0]

12 Aug 2020

SUMMARY OF MAIN CHANGES FOR PAPER

Paper Title: The development of an alternative growth chart for estimated fetal weight in the absence of ultrasound: Application in Indonesia

Manuscript ID: PONE-D-19-33469

Response to reviewers' comments 

The authors would like to thank the editor and the reviewers for their constructive comments on the paper. Responses to the editor's and reviewers' comments have been provided immediately after the comment.

Additional requirements

"When submitting your revision, we need you to address these additional requirements.

and

Response to additional requirements 1: 

Thank you for the feedback. We have revised the manuscript based on PLOS ONE's style requirements and templates, including file naming.

a) Did participants provide their written or verbal informed consent to participate in this study?

b) If consent was verbal, please explain i) why written consent was not obtained, ii) how you documented participant consent, and iii) whether the ethics committees/IRB approved this consent procedure".

Response to additional requirements 2: 

We have amended the ethics statement as suggested and highlighted the changes in lines 190-207 in the revised manuscript.

Additional Editor Comments

"Please revise the manuscript according to the Author's instructions and Manuscript preparation guidelines."

Response to additional Editor Comments: 

We have revised the manuscript according to the Author's instructions and Manuscript preparation guidelines.

Reviewer 1 

Comment 1: 

"No, I don't have any comments for authors. But he did not follow the Author's instructions and manuscript preparation guidelines. The author needs to follow the journal's manuscript preparation guidelines to create a good manuscript".

Authors' response to comment 1:

Thank you for the suggestion. We have revised the manuscript and followed the Author's instructions and Manuscript preparation guidelines

Reviewer 2

Major Revisions 

Comment 1: 

"In results, while describing the general characteristics of the study population, the total number of enrolled women is missing, and without this, it is difficult to understand various percentages given in this section."

Authors' response to comment 1:

Thank you for the constructive feedback. We have put the total number of enrolled women used for: the local evaluation, the validation of the prediction models deployed in the development of existing growth charts for estimated fetal weight, and the comparison with the proposed model. The correction is highlighted on line 250 in the revised manuscript.

Comment 2: 

"While describing tables 1 and 2, authors have reported all estimates (ME, MPE, MAPE, PEDPE, etc.) in lines from 285 to 297. This text should be rephrased to simplify the message of both tables".

Authors responses to comments 2:

Thank you for the suggestion. We have rephrased the paragraph to simplify the message of Tables 1 and 2. The changes are highlighted in lines 269-274 in the revised manuscript. 

Comment 3:

"In Tables 1, all median percentages and median absolute percentages for various models are the same, whereas, in table 2, both values are different for all models except Stirnemann (2017). These should be rechecked".

Authors responses to comments 3:

Thank you for the critical feedback. We have rechecked, and all median percentages and median absolute percentages for various models in both Tables 1 and 2 are correct.

Comment 4:

"The lines 320 and 321 regarding F test should be removed from here as it has already been described in lines 134 – 136 in methods section". 

Authors responses to comments 4:

Thank you for the correction. We have removed the sentence in lines 320-321 regarding the F test in the revised manuscript.

Comment 5:

"Lines 373 and 374 "However, since ………. Problems." Should be part of the discussion". 

Authors responses to comments 5:

Thank you for the suggestion. We have removed the sentence in the revised manuscript.

Comment 6:

"Lines 413 and 414 on page 19 regarding F test should be moved to methods".

Authors responses to comments 6:

Thank you for the constructive feedback. We have moved lines 413 and 414 to the Methods section and highlighted in lines 133-135 in the revised manuscript. 

Comment 7:

"Lines 418 – 420 can be moved to discussion/conclusion. Similarly, lines 457 – 459 on page 21, "The reason ………….EFW-FH model" should be moved to the discussion."

Authors responses to comments 7:

Thank you for the critical feedback. We have moved lines 418-420 and lines 457-459 to the Discussion section. The changes are highlighted in lines 524-525 and lines 577-579, respectively, in the revised manuscript. 

Comment 8:

"The text "Since the aim ………………percentiles" in lines 501 and 502 should be the part of the discussion".

Authors responses to comments 8:

Thank you for the constructive feedback. We have moved lines 501-502 to the Discussion section. The changes are highlighted in lines 621-622 in the revised manuscript.

Comment 9: 

"The conclusion part should be revised as most of the text given in the 1st and 3rd para of conclusions should be part of the discussion. Similarly, the strengths and limitations of the study given on pages 32 and 33 under the conclusion section should be moved to the discussion section".

Authors responses to comments 9:

Thank you for the comment and suggestions. We have moved the 1st and 3rd paragraphs of the conclusion to the Discussion section. The changes are highlighted in lines 564-571 and lines 591-600, respectively in the revised manuscript. 

We also moved the strengths and limitations of the study to the Discussion section. The changes are also highlighted in lines 627-631 and lines 632-635 in the revised manuscript.

Minor Corrections

Comment 10: 

"The abstract can be presented with standard headings, such as Introduction, methods, results, and Conclusion." 

Authors responses to comments 10:

Thank you for the comment. However, the abstract has been presented and written based on the following PLOS ONE's style templates https://journals.plos.org/plosone/s/file?id=wjVg/PLOSOne_formatting_sample_main_body.pdf. 

Comment 11: 

"Some language corrections will further improve the quality of the paper." 

Authors responses to comments 11:

Thank you for the suggestion. This paper has been proofread and corrected from some language errors.

---

## [Decision Letter · Decision Letter 1]

28 Sep 2020

The development of an alternative growth chart for estimated fetal weight in the absence of ultrasound: application in Indonesia

PONE-D-19-33469R1

Dear Dr. Anggraini,

We’re pleased to inform you that your manuscript has been judged scientifically suitable for publication and will be formally accepted for publication once it meets all outstanding technical requirements.

Kind regards,

Giovanni Delli Carpini

Academic Editor

PLOS ONE

Additional Editor Comments (optional):

Reviewers' comments:

Reviewer's Responses to Questions

**Comments to the Author**

1. If the authors have adequately addressed your comments raised in a previous round of review and you feel that this manuscript is now acceptable for publication, you may indicate that here to bypass the “Comments to the Author” section, enter your conflict of interest statement in the “Confidential to Editor” section, and submit your "Accept" recommendation.

Reviewer #1: All comments have been addressed

Reviewer #2: All comments have been addressed

2. Is the manuscript technically sound, and do the data support the conclusions?

Reviewer #1: Yes

Reviewer #2: (No Response)

3. Has the statistical analysis been performed appropriately and rigorously? 

Reviewer #1: Yes

Reviewer #2: (No Response)

4. Have the authors made all data underlying the findings in their manuscript fully available?

Reviewer #1: Yes

Reviewer #2: (No Response)

5. Is the manuscript presented in an intelligible fashion and written in standard English?

Reviewer #1: Yes

Reviewer #2: (No Response)

6. Review Comments to the Author

Reviewer #1: (No Response)

Reviewer #2: (No Response)

7. PLOS authors have the option to publish the peer review history of their article (what does this mean?). If published, this will include your full peer review and any attached files.

Reviewer #1: No

Reviewer #2: No

---

## [Editor Report · Acceptance letter]

2 Oct 2020

PONE-D-19-33469R1 

The development of an alternative growth chart for estimated fetal weight in the absence of ultrasound: Application in Indonesia 

Dear Dr. Anggraini:

I'm pleased to inform you that your manuscript has been deemed suitable for publication in PLOS ONE. Congratulations! Your manuscript is now with our production department. 

Kind regards, 

on behalf of

Dr. Giovanni Delli Carpini 

Academic Editor

PLOS ONE